# Public Awareness of the Association between Periodontal Disease and Systemic Disease

**DOI:** 10.3390/healthcare11010088

**Published:** 2022-12-28

**Authors:** Fahd Alsalleeh, Abdulmalik S. Alhadlaq, Nora A. Althumiri, Norah AlMousa, Nasser F. BinDhim

**Affiliations:** 1Department of Restorative Dental Sciences, College of Dentistry, King Saud University, Riyadh 11545, Saudi Arabia; 2Department of Infectious Diseases, King Saud Medical City, Riyadh 12746, Saudi Arabia; 3Sharik Association for Health Research, Riyadh 13326, Saudi Arabia; 4College of Medicine, Alfaisal University, Riyadh 11533, Saudi Arabia

**Keywords:** diabetes mellitus, coronary heart disease, atherosclerosis, hypertension, periodontal disease

## Abstract

Periodontal disease is associated with other non-communicable diseases including diabetes mellitus, coronary heart disease and atherosclerosis, hypertension, and respiratory tract infections. This association merits careful study of the general population’s awareness level in order to leverage the current state of science to improve general health and quality of life. This study included 502 residents of Saudi Arabia who received computer-assisted interviews to fill up the survey. Results indicated a low level of awareness among the study population regarding the association of periodontal disease to diabetes mellitus, coronary heart disease and atherosclerosis, hypertension, and respiratory tract infections. A higher level of awareness was noticed with individuals with periodontal disease, themselves or a member of their family having a systemic disease, and who have a specialized person or scientific article as their source of information. This observed low level of awareness deserves the attention of public health authorities to prioritize programs that increase the awareness, improve health, and reduce burden of systemic diseases of high prevalence, morbidity, and mortality.

## 1. Introduction

Periodontal disease is a chronic inflammatory infection that results in the progressive destruction of supporting tissues of the teeth, with continuing loss of connective tissue attachment and resorption of the bone [1]. According to the World Health Organization (WHO), severe periodontal disease affects over one billion individuals, constituting about fourteen percent of the world population [2]. Different risk factors are associated with periodontal disease, the strongest being cigarette smoking and diabetes mellitus [3]. If left untreated, periodontal disease can result in patients with missing teeth, affecting their well-being and quality of life [4,5,6].

Periodontal disease appears to be associated with other non-communicable systemic diseases. Not only is periodontitis a risk for difficult glycemic control, but it is also a complication of diabetes mellitus, resulting in a "bidirectional association" between them [7]. Periodontal disease affects glycemic control and increases the risk of developing hyperglycemia, which increases with periodontal disease severity [8,9,10,11]. Some evidence suggests that the treatment of periodontal disease would result in better glycemic control [12]. For coronary heart disease and atherosclerosis, patients with periodontal disease are at an increased risk for these diseases, including acute myocardial infarctions [13]. In addition, oral bacteria have been detected in atherosclerotic plaques [5,14]. Periodontal disease has been reported to affect hypertension, causing increased systolic and diastolic blood pressure [15]. Moreover, other reports have demonstrated systolic and diastolic blood pressure reduction following periodontal treatment [16]. For respiratory diseases, it has been suggested that the aspiration of pathogenic oral bacteria found in a poorly maintained oral cavity might result in pneumonia development or chronic obstructive pulmonary disease exacerbations [5,17,18].

Awareness of the general public about the association of periodontal disease with non-communicable systemic diseases can encourage individuals to seek proper dental care to prevent and treat existing dental conditions, particularly periodontal disease, thereby reducing the risk of non-communicable systemic diseases and contributing to the improvement of some existing non-communicable systemic disease conditions [5,19]. However, while healthcare providers have a high level of awareness regarding the association between periodontal disease and systemic disease [20,21,22,23], patients, on the other hand, have low levels of awareness regarding this link [24].

Limited evidence in the literature is available addressing the general public’s awareness regarding the connection between oral and systemic diseases. Therefore, the present study aimed to assess the general public’s awareness regarding the connection between periodontal disease and non-communicable systemic diseases, namely, diabetes mellitus, coronary heart disease and atherosclerosis, hypertension, and respiratory disease.

## 2. Materials and Methods

### 2.1. Study Design

A nationwide cross-sectional survey was conducted in Saudi Arabia via computer-assisted phone interviews between September and October 2022.

### 2.2. Sampling and Sample Size

A proportionate quota-sampling procedure was used to obtain an approximately balanced allocation of participants based on sex across the five main regions of Saudi Arabia (the central region, western region, eastern region, northern region, and southern region), leading to a total of ten quotas. The sample size was generated using a medium effect size, at 0.30, with an 80% sampling power and a 95% confidence interval. Therefore, 50 participants were needed in each quota, and a total sample size of 500 participants was required in this research project. This study used “ZdataCloud” (Informed Decision Making (IDM), Version 2, Riyadh, Saudi Arabia), an electronic research data collection and management platform that supports computer-assisted phone interviews, which had an integrated eligibility and sampling modules, to control the sample’s distribution and to prevent human-related sampling bias [25]. Two factors, sex and region, were used to determine the adherence to the sampling quota.

### 2.3. Participant Recruitment

Saudi residents and Arabic-speaking adults (≥18 years old) were invited to participate according to the protocol approved by the King Saud University Institutional Review Board (IRB; Research Project No. 22/0523/IRB 3 July 2022). A random phone number list from the Sharik Association for Health Research (Sharik research participants’ database) to identify prospective participants was utilized [26]. The Sharik research participants’ database includes individuals who provided consent and are willing to participate in future studies. The database includes more than 172,000 potential research participants distributed across all regions of Saudi Arabia and continues to expand [26]. Potential participants were called up to three times. If there was no response, the number of a new participant with similar demographical characteristics was generated from the Sharik database. After explaining the study to the potential participant, consent to participate was obtained, and the interviewer assessed the eligibility of each participant on the data collection platform based on the sex and region quota completion criteria. Once the sampling quota was complete, recruitment ceased automatically [25]. 

### 2.4. Survey Design and Outcome Measures

The survey instrument consisted of a five-point Likert scale (strongly disagree, disagree, I do not know, agree, and strongly agree) to assess awareness of the general public regarding the association of having periodontal disease with four systemic diseases, namely diabetes mellitus, coronary heart disease and atherosclerosis, hypertension, and respiratory tract infections. In addition, demographic information of the participants, including sex, age, region, education level, and frequency of dental visits, was collected. Furthermore, self-perceived/reported affliction by dental disease (untreated dental caries or periodontal disease) or systemic diseases of interest in this study for self or family members and the source of their stated selection regarding the association between periodontal disease and systemic disease, has been collected. The survey had ten close-ended questions.

We conducted several rounds of linguistic validation to ensure clarity and understanding of the survey questions via focus groups. In addition, group members were asked to review and discuss survey questions and answers. Based on the outcomes of linguistic validation procedures, the survey was further edited, and the final version was approved.

### 2.5. Data Analysis

Descriptive analysis was used to describe the variables. In addition, awareness about the association of periodontal disease with systemic disease was described as frequencies and percentages within each demographic item, self-reported dental and health condition, and source of information.

Inferential statistical analysis was performed to evaluate the relationship between various socio-demographic factors and the perceived association of periodontal disease with systemic disease. The relationship between various socio-demographic factors and the perceived association of periodontal disease with systemic disease were examined using a chi-square test or Fisher’s Exact test, depending on data frequency. The perceived association data were analyzed using a dichotomized version of the Likert scale (agree and strongly agree combined). Significant factors from bivariate analysis were further explored with multivariate analysis using binary logistic regression models.

Level of significance 0.05 was used for all inferential analysis, with *p*-values < 0.05 reported as statistically significant. Both descriptive and inferential analysis was performed using SPSS Statistics software (IBM Corp, Version 28.0. Armonk, NY, USA).

## 3. Results

Out of 808 potential participants contacted, 502 accepted to participate and completed the interview (response rate 62.1%). The condition was satisfied regarding equal sex and region distribution of the sample (Table 1).

Almost half of the sample (48%) believe that they are free of dental caries and periodontal disease, while 74.8% believe that they are free of the systemic diseases studied, and 37.6% have no family member affected by diabetes mellitus, coronary heart disease and atherosclerosis, hypertension, or respiratory tract infection (Table 2).

In the overall sample, most participants chose “I do not know” in the association between periodontal disease and systemic disease (Table 3). Diabetes mellitus was the disease with the least “I do not know” response (47.2%), hypertension and respiratory tract infection were the most with 54.0% each, and coronary heart disease and atherosclerosis was 51.1% (Table 3).

The highest percentage of respondents that believe there is an association between periodontal disease and systemic disease (agree and strongly agree) was for diabetes mellitus (35.7%), followed by hypertension (28.1%). Most participants (59.8%) selected the association between periodontal disease and systemic disease based on their opinion (Table 4).

### 3.1. Periodontal Disease and Diabetes Mellitus

The general population’s awareness regarding the association of periodontal disease and diabetes mellitus was demonstrated as the response to the association of periodontal disease with diabetes mellitus, categorized according to demographic, dental and health status, and source of information. Descriptive analysis using frequencies and percentages is presented in Table 5.

The strength of the association between periodontal disease and diabetes mellitus was examined between levels of various factors, and results are presented in Table 6. The analysis was performed using a chi-square test or Fisher’s Exact test.

The analysis of the perceived strength of the association of periodontal disease with diabetes showed several statistically significant predictors. These factors result in an increased proportion of responses who agree or strongly agree with the statement of awareness of periodontal disease and diabetes mellitus association. Here are the factors that are statistically significant:Participant having untreated periodontal disease (*p* < 0.001)Participant having diabetes mellitus (*p* = 0.004)Member(s) of participant’s family having coronary heart disease and atherosclerosis (*p* = 0.031)Participant having an information source as a specialized person in media or on social media (*p* = 0.002 and *p* < 0.001, respectively)Having an information source as scientific article (*p* < 0.001)

The significant predictors from bivariate analysis were entered into multivariate analysis–binary logistic regression model (Table 7). The dependent variable was the perceived association between periodontal disease and diabetes mellitus, with two levels: strongly agree/agree vs. strongly disagree/disagree/neutral. The results of multi-variate analysis suggest that only three factors remain statistically significant:Participant having untreated periodontal disease (OR = 1.96, *p* = 0.007)Having an information source as a specialized person on social media (OR = 2.34, *p* = 0.013)Having an information source as scientific article (OR = 2.98, *p* = 0.008)

### 3.2. Periodontal Disease and Coronary Heart Disease and Atherosclerosis

The general population’s awareness regarding the association of periodontal disease and coronary heart disease and atherosclerosis was demonstrated as the response to the association of periodontal disease with coronary heart disease and atherosclerosis, categorized according to demographic, dental and health status, and source of information. Descriptive analysis using frequencies and percentages is presented in Table 8.

The strength of the association between periodontal disease and coronary heart disease and atherosclerosis was examined between levels of various factors, and results are presented in Table 9. The analysis was performed using a chi-square test or Fisher’s Exact test.

The analysis of the perceived strength of the association of periodontal disease with coronary heart disease and atherosclerosis showed several statistically significant predictors. Here are the significant factors that result in an increased proportion of responses who agree or strongly agree with the statement of awareness of periodontal disease and coronary heart disease and atherosclerosis association:Participant having untreated periodontal disease (*p* = 0.006)Participant having coronary heart disease and atherosclerosis (*p* = 0.010)Member(s) of participant’s family having coronary heart disease and atherosclerosis (*p* = 0.010)Member(s) of participant’s family not having any systemic diseases (*p* = 0.033)Having an information source as a specialized person in media or on social media (*p* < 0.001)Having an information source as general accounts on social media (*p* < 0.001)Having an information source as scientific article (*p* < 0.001)

The significant predictors from bivariate analysis were entered into multivariate analysis–binary logistic regression model (Table 10). The dependent variable was the perceived association between periodontal disease and coronary heart disease and atherosclerosis, with two levels: strongly agree/agree vs. strongly disagree/disagree/neutral. The results of multivariate analysis suggest that only seven factors remain statistically significant:Participant having untreated periodontal disease (OR = 2.13, *p* = 0.010)Participant having coronary heart disease and atherosclerosis (OR = 2.70, *p* = 0.004)Member(s) of participant’s family having diabetes mellitus (OR = 0.28, *p* < 0.001)Member(s) of participant’s family having coronary heart disease and atherosclerosis (OR = 2.79, *p* = 0.003)Having an information source as a specialized person on social media (OR = 2.98, *p* = 0.003)Having an information source as general accounts on social media (OR = 2.76, *p* = 0.004)Having an information source as scientific article (OR = 6.43, *p* < 0.001)

### 3.3. Periodontal Disease and Hypertension

The general population’s awareness regarding the association of periodontal disease and hypertension was demonstrated as the response to the association of periodontal disease with hypertension, categorized according to demographic, dental and health status, and source of information. Descriptive analysis using frequencies and percentages is presented in Table 11.

The strength of the association between periodontal disease and hypertension was examined between levels of various factors, and results are presented in Table 12. The analysis was performed using a chi-square test or Fisher’s Exact test.

The analysis of the perceived strength of the association of periodontal disease with hypertension showed several statistically significant predictors. Here are the significant factors that result in an increased proportion of responses who agree or strongly agree with the statement of awareness of periodontal disease and hypertension association:Participant having untreated periodontal disease (*p* < 0.001)Participant having hypertension (*p* = 0.017)Member(s) of participant’s family having coronary heart disease and atherosclerosis (*p* = 0.013)Having an information source as a specialized person in media or on social media (*p* = 0.004 and *p* < 0.001, respectively)Having an information source as general accounts on social media (*p* < 0.001)Having an information source as scientific article (*p* < 0.001)

The following factors significantly decrease the proportion of responses who agree or strongly agree with the statement of awareness of periodontal disease and hypertension association:Member(s) of participant’s family having diabetes mellitus (*p* = 0.045)Having respondent’s own opinion as an information source (*p* = 0.023)

The significant predictors from bivariate analysis were entered into multivariate analysis–binary logistic regression model (Table 13). The dependent variable was the perceived association between periodontal disease and hypertension, with two levels: strongly agree/agree vs. strongly disagree/disagree/neutral. The results of multivariate analysis suggest that only six factors remain statistically significant:Participant having untreated periodontal disease (OR = 2.42, *p* < 0.001)Member(s) of participant’s family having diabetes mellitus (OR = 0.55, *p* = 0.009)Member(s) of participant’s family having coronary heart disease and atherosclerosis (OR = 1.99, *p* = 0.018)Having an information source as a specialized person on social media (OR = 2.14, *p* = 0.031)Having an information source as general accounts on social media (OR = 2.71, *p* = 0.002)Having an information source as scientific article (OR = 3.15, *p* = 0.007)

### 3.4. Periodontal Disease and Respiratory Tract Infections

The general population’s awareness regarding the association of periodontal disease and respiratory tract infections was demonstrated as the response to the association of periodontal disease with respiratory tract infections, categorized according to demographic, dental and health status, and source of information. Descriptive analysis using frequencies and percentages is presented in Table 14.

The strength of the association between periodontal disease and respiratory tract infections was examined between levels of various factors, and results are presented in Table 15. The analysis was performed using a chi-square test or Fisher’s Exact test.

The analysis of the perceived strength of the association of periodontal disease with respiratory tract infections showed several statistically significant predictors. Here are the significant factors that result in an increased proportion of responses who agree or strongly agree with the statement of awareness of periodontal disease and respiratory tract infections association:Member(s) of participant’s family having no systemic diseases (*p* = 0.026)Having an information source as a specialized person in media or on social media (*p* = 0.029 and *p* = 0.006, respectively)Having an information source as general accounts on social media (*p* < 0.001)Having an information source as scientific article (*p* = 0.005)Participant being in the health field (*p* = 0.003)

The following factors significantly decrease the proportion of responses who agree or strongly agree with the statement of awareness of periodontal disease and respiratory tract infections association:Member(s) of participant’s family having diabetes mellitus (*p* < 0.001)Having respondent’s own opinion as an information source (*p* = 0.007)

The significant predictors from bivariate analysis were entered into multivariate analysis–binary logistic regression model (Table 16). The dependent variable was the perceived association between periodontal disease and respiratory tract infections, with two levels: strongly agree/agree vs. strongly disagree/disagree/neutral. The results of multivariate analysis suggest that only four factors remain statistically significant:Member(s) of participant’s family having diabetes mellitus (OR = 0.45, *p* = 0.012)Having an information source as general accounts on social media (OR = 2.99, *p* = 0.001)Having an information source as scientific article (OR = 2.62, *p* = 0.026)Participant being in the health field (OR = 3.71, *p* = 0.002)

## 4. Discussion

This study investigated the general public’s awareness regarding the association of periodontal disease with four systemic diseases: diabetes mellitus, coronary heart disease and atherosclerosis, hypertension, and respiratory tract infection. The overall results indicated a low level of awareness, where about 50% of individuals did not know about the association. However, when the data were analyzed according to various demographic, dental and medical health, and source of information variables, differences among the groups started to emerge. In general, people affected with periodontal disease, affected with the same systemic disease, have a family member with coronary heart disease and atherosclerosis, or use scientific articles or specialized individuals as a source of information will tend to be more aware of the association between having periodontal disease and the incidence or severity of diabetes mellitus, coronary heart disease and atherosclerosis, hypertension, and respiratory tract infection.

The low level of awareness about the association between periodontal disease and systemic disease reported herein agrees with previous studies [24]. For example, the level of awareness about the effect of periodontal disease on diabetes mellitus in this study (35.7%) is similar to the average of the previously reported 21.8% for Saudi diabetic patients [27], and 44% for German patients [28]. In addition, the low level of awareness for the association between periodontal disease and coronary heart disease and atherosclerosis in this study (21.9%) agrees with a study that showed it to be less than 10% [29]. Despite the limitations of this study, namely being a cross-sectional study, self-reporting of dental and medical conditions by participants, and involving only Saudi residents, the general trend of this work’s findings is similar to other studies. However, the variability in the extent of awareness between the current study and the other studies could be attributed to the differences in study samples and study methodology, as well as to the difference in the time of study conduction, where the reported association between periodontal disease and systemic disease increases with time, thereby increasing the level of awareness is expected. 

The results revealed that individuals with periodontal and systemic diseases had increased awareness of their association. It is reasonable to hypothesize that such awareness had come from their healthcare providers, who have a high level of awareness regarding the association between periodontal disease and systemic disease [21,22]. In addition, findings of this study (Table 5, Table 6, Table 7, Table 8, Table 9, Table 10, Table 11, Table 12, Table 13, Table 14, Table 15 and Table 16) shows that participants’ information from specialized individuals contributed to the increased awareness and may reflect a change in society’s learning behavior, where active members increasingly influence the general public on conventional and social media. Furthermore, higher awareness of the association demonstrated by subjects in this study who gained information from scientific articles may reflect a shift in knowledge dissemination norms brought by the ease of reach to open access journals on the internet.

The association of periodontal disease and systemic disease is an active area of research, with evidence suggesting a unidirectional, only one disease affecting the other, or bidirectional relationship. Each disease will adversely affect the other in this relationship. Periodontal disease association with diabetes mellitus is probably the most established bidirectional relationship, with periodontal disease affecting glycemic control [8], and individuals with periodontal disease have 19–33% more risk of developing hyperglycemia [9,10], which is increased with periodontal disease severity [11]. On the other hand, having diabetes mellitus is associated with three-times increased risk of having periodontal disease [30]. Individuals with severe periodontal disease have 11% higher risk of acute myocardial infarction [13]. In addition, tooth loss, a common end-stage result of periodontal disease, is associated with a 3% increased risk of coronary heart disease [31]. Periodontal disease-affected individuals have an increased mean systolic (3.36 mm Hg) and diastolic (2.16 mm Hg) blood pressure compared to healthy controls [15]. In addition, periodontal disease was associated with increased odds (2.3) of being diagnosed with hyper-tension (systolic blood pressure > 140 mm Hg) [15]. Interestingly, six months after periodontal disease treatment, there was a reduction in the mean systolic and diastolic blood pressure by 12.57 mm Hg and 9.65 mm Hg, respectively [16]. However, the evidence supporting blood pressure reduction following periodontal disease therapy is inconclusive [32]. Periodontal disease increases the risk of pneumonia, with an odds ratio of up to 4.4 reported in one study [17,18]. This association between periodontal disease and systemic disease merits efforts concerted at improving the periodontal health of the general public, both afflicted with systemic diseases and healthy individuals. In addition to the improved quality of life associated with having healthy teeth [6], it could also contribute to preventing or reducing the severity of some systemic diseases [5,19].

Improving oral health requires multi-level actions, starting with increasing the awareness of the general public regarding the association of periodontal disease and systemic disease utilizing the most effective means. Although members of medical and dental health teams have a high level of awareness regarding periodontal disease association with systemic diseases [20,21,22], this knowledge is yet to be transferred effectively to patients, as demonstrated in this and previous studies [24]. It seems that a lack of time, inadequate knowledge and training in oral health, as well as a lack of interaction with oral health care providers are the main barriers to providing patient education about the association between oral and systemic disease [21]. Therefore, efforts should be made to integrate patient education regarding the association between periodontal and systemic disease within care delivery; perhaps a support medical staff can be dedicated to such a task. In addition, medical care should include dental examination and treatment as part of systemic disease care through communication and collaboration with oral health care professionals [33,34]. Finally, the most effective venue to increase awareness for the general public should be utilized. The findings of this study show that specialized individuals, in various traditional and new social media outlets, as the source of high-awareness responses might be a good starting point.

## 5. Conclusions

Awareness of the general public regarding the association between periodontal disease and systemic disease is low. However, individuals having periodontal disease, being affected with the same systemic disease, having a family member with coronary heart disease and atherosclerosis, or using scientific articles or specialized individuals as a source of information were more aware of the association between having periodontal disease and the incidence or severity of diabetes mellitus, coronary heart disease and atherosclerosis, hypertension, and respiratory tract infection. This increased awareness of certain segments of the study population about the association of periodontal disease and systemic disease should form a basis to carefully understand the effect of various factors and then implement mechanisms to augment the awareness of the general population.

## Figures and Tables

**Table 1 healthcare-11-00088-t001:** Demographic characteristics of the study participants.

Characteristic	Number	Percentage
Sex		
Male	250	49.8%
Female	252	50.2%
Age group		
18–24	93	18.5%
25–44	259	51.6%
45–64	138	27.5%
>65	12	2.4%
Region distribution		
Southern region	101	20.1%
Eastern region	100	19.9%
Northern region	100	19.9%
Central region	100	19.9%
Western region	101	20.1%
Education level		
Elementary school	32	6.4%
Middle school	33	6.6%
High school	107	21.3%
Post-secondary diploma	61	12.2%
Bachelor’s degree	239	47.6%
Master’s degree	20	4.0%
Ph.D.	10	2.0%
Frequency of dental visits		
Once every six months	68	13.5%
Once a year	57	11.4%
Only when I need dental treatment	184	36.7%
Only when I feel pain	162	32.3%
I never visit the dentist	31	6.2%

**Table 2 healthcare-11-00088-t002:** Dental and systemic health conditions affecting the study participants.

Variable	Number	Percentage *
Dental health		
Untreated dental caries	219	43.6%
Untreated periodontal disease	88	17.5%
No dental caries or periodontal disease	243	48.0%
Systemic disease affecting the participant		
Diabetes mellitus	64	12.7%
Coronary heart disease and atherosclerosis	26	5.2%
Hypertension	70	13.9%
Respiratory tract infections	14	2.8%
Not having any of the above diseases	376	74.8%
Systemic disease affecting the participant’s family member(s)		
Diabetes mellitus	274	54.5%
Coronary heart disease and atherosclerosis	75	14.9%
Hypertension	201	40.0%
Respiratory tract infections	38	7.6%
Not having any of the above diseases	189	37.6%

* The totals of percentages in each category are more than 100% due to individuals having more than one disease.

**Table 3 healthcare-11-00088-t003:** Awareness of the study participants about the association between periodontal disease and systemic disease.

Awareness Question	Strongly Disagree	Disagree	I Do not Know	Agree	Strongly Agree
	*n* (%)	*n* (%)	*n* (%)	*n* (%)	*n* (%)
Do you believe that periodontal disease is associated with the following systemic diseases? (is having periodontal disease associated with increased incidence/severity of these diseases?):					
Diabetes mellitus	21 (4.2%)	65 (12.9%)	237 (47.2%)	137 (27.3%)	42 (8.4%)
Coronary heart disease and atherosclerosis	19 (3.8%)	113 (22.5%)	260 (51.8%)	82 (16.3%)	28 (5.6%)
Hypertension	23 (4.6%)	67 (13.3%)	271 (54.0%)	105 (20.9%)	35 (7.2%)
Respiratory tract infections	23 (4.6%)	99 (19.7%)	271 (54.0%)	82 (16.3%)	27 (5,4%)

Percentages are calculated for each row.

**Table 4 healthcare-11-00088-t004:** Source of information for the association between periodontal disease and systemic disease.

Variable	Number	Percentage *
Specialized person appearing in written, visual, or audio media	29	5.8%
Post on social media by a specialized person	45	9.0%
General accounts on social media	56	11.2%
Scientific article	31	6.2%
My opinion	300	59.8%
I am in the dental field	17	3.4%
I am in the health field	30	6.0%
Personal message of a social group (for example WhatsApp, Telegram, …)	39	7.8%

* The total of percentage is more than 100% due to individuals having more than one source of information.

**Table 5 healthcare-11-00088-t005:** Awareness about the association between periodontal disease and diabetes mellitus distributed by various factors.

Variable	Strongly Disagree		Disagree		I Do not Know		Agree		Strongly Agree	
	*n*	(%)	*n*	(%)	*n*	(%)	*n*	(%)	*n*	(%)
Sex										
Male	11	4.4%	33	13.2	123	49.2	60	24%	23	9.2%
Female	10	4.0%	32	12.7%	114	45.2%	77	30.6%	19	7.5%
Age group										
18–24	4	4.3%	16	17.2%	40	43.0%	28	30.1%	5	5.4%
25–44	10	3.9%	35	13.5%	123	47.5%	67	25%	24	9.3%
45–64	6	4.3%	13	9.4%	69	50.0%	39	28.3%	11	8.0%
>65	1	8.3%	1	8.3%	5	41.7%	3	25.0%	2	16.7%
Education level										
Elementary school	0	0%	8	25%	13	40.6%	7	21.9%	4	12.5%
Middle school	2	6.1%	2	6.1%	16	48.5%	8	24.2%	5	15.2%
High school	3	2.8%	16	15%	46	43%	33	30.8%	9	8.4%
Post-secondary diploma	5	8.2%	4	6.6%	33	54.1%	12	19.7%	7	11.5%
Bachelor’s degree	11	4.6%	32	13.4%	115	48.1%	68	28.5%	13	5.4%
Master’s degree	0	0%	2	10%	9	45%	8	40.0%	1	5.0%
Ph.D.	0	0%	1	10%	5	50%	1	10.0%	3	30.0%
Frequency of dental visits										
Once every six months	4	5.9%	9	13.2%	27	39.7%	21	30.9%	7	10.3%
Once a year	0	0.0%	8	14.0%	30	52.6%	14	24.6%	5	8.8%
Only when I need dental treatment	6	3.3%	24	13.0%	83	45.1%	51	27.7%	20	10.9%
Only when I feel pain	8	4.9%	22	13.6%	78	48.1%	46	28.4%	8	4.9%
I never visit the dentist	3	9.7%	2	6.5%	19	61.3%	5	16.1%	2	6.5%
Dental health										
Untreated dental caries	11	5.0%	21	9.5%	111	50.2%	66	29.9%	12	5.4%
Untreated periodontal disease	2	2.3%	12	13.6%	29	33.0%	36	40.9%	9	10.2%
No dental caries or periodontal disease	9	3.7%	38	15.8%	112	45.5%	56	23.2%	26	10.8%
Systemic disease affecting the participant										
Diabetes mellitus	4	6.3%	11	17.2%	16	25%	26	40.6%	7	10.9%
Coronary heart disease and atherosclerosis	2	7.7%	3	11.5%	8	30.8%	11	42.3%	2	7.7%
Hypertension	5	7.1%	9	12.9%	28	40.0%	24	34.3%	4	5.7%
Respiratory tract infections	2	14.3%	3	21.4%	5	35.7%	4	28.6%	0	0.0%
Not having any of the above diseases	11	2.9%	49	13.0%	191	50.8%	93	24.7%	32	8.5%
Systemic disease affecting the participant’s family member(s)										
Diabetes mellitus	16	7.0%	35	15.4%	104	45.6%	59	25.9%	14	6.1%
Coronary heart disease and atherosclerosis	4	5.3%	9	12.0%	27	36.0%	25	33.3%	10	13.3%
Hypertension	14	7.0%	26	12.9%	93	46.3%	53	26.4%	15	7.5%
Respiratory tract infections	1	2.6%	8	21.1%	17	44.7%	11	28.9%	1	2.6%
Not having any of the above diseases	2	1.1%	20	10.6%	95	50.3%	51	27.0%	21	11.1%
Source of information										
Specialized person appearing in written, visual, or audio media	4	13.8%	1	3.4%	6	20.7%	15	51.7%	3	10.3%
Post on social media by a specialized person	2	4.4%	4	8.9%	12	26.7%	19	42.2%	8	17.8%
General accounts on social media	7	10.7%	7	12.5%	19	33.9%	17	30.4%	7	12.5%
Scientific article	2	6.5%	0	0.0%	9	29.0%	17	54.8%	3	9.7%
My opinion	7	2.3%	36	12.0%	156	52.0%	86	28.7%	15	5.0%
I am in the dental field	0	0.0%	1	11.8%	9	52.9%	5	29.4%	1	5.9%
I am in the health field	3	10.0%	5	16.7%	7	23.3%	10	33.3%	5	16.7%
Personal message of a social group (for example WhatsApp, Telegram, …)	3	7.7%	9	23.1%	12	30.8%	11	28.2%	4	10.3%

The totals of percentages are more than 100% due to individuals having more than one condition simultaneously. Per-centages are calculated for each row.

**Table 6 healthcare-11-00088-t006:** Awareness about the association between periodontal disease and diabetes mellitus distributed by various factors.

Variable	N (%) of Strongly Disagree, Disagree, Neutral	N (%) of Agree and Strongly Agree	Comparison Test ^1^
Sex Male Female	167 (66.8%) 156 (61.9%)	83 (33.2%) 96 (38.1%)	*p* = 0.252 ^C^
Age group 18–24 25–44 45–64 >65	60 (64.5%) 168 (64.9%) 88 (63.8%) 7 (58.3%)	33 (35.5%) 91 (35.1%) 50 (36.2%) 5 (41.7%)	*p* = 0.971 ^C^
Education level Elementary school Middle school High school Post-secondary diploma Bachelor’s degree Master’s degree Ph.D.	21 (65.6%) 20 (60.6%) 65 (60.7%) 42 (68.9%) 158 (66.1%) 11 (55.0%) 6 (60.0%)	11 (34.4%) 13 (39.4%) 42 (39.3%) 19 (31.1%) 81 (33.9%) 9 (45.0%) 4 (40.0%)	*p* = 0.864 ^C^
Frequency of dental visits Once every six months Once a year Only when I need dental treatment Only when I feel pain I never visit the dentist	40 (58.8%) 38 (66.7%) 113 (61.4%) 108 (66.7%) 24 (77.4%)	28 (41.2%) 19 (33.3%) 71 (38.6%) 54 (33.3%) 7 (22.6%)	*p* = 0.352 ^C^
Dental health Untreated dental caries Untreated periodontal disease No dental caries or periodontal disease	143 (64.7%) 43 (48.9%) 159 (66.0%)	78 (35.3%) 45 (51.1%) 82 (34.0%)	*p* = 0.880 ^C^ *p* < 0.001 ^C^ *p* = 0.463 ^C^
Systemic disease affecting the participant Diabetes mellitus Coronary heart disease and atherosclerosis Hypertension Respiratory tract infections Not having any of the above diseases	31 (48.4%) 13 (50.0%) 42 (60.0%) 10 (71.4%) 251 (66.8%)	33 (51.6%) 13 (50.0%) 28 (40.0%) 4 (28.6%) 125 (33.2%)	*p* = 0.004 ^C^ *p* = 0.117 ^C^ *p* = 0.414 ^C^ *p* = 0.575 ^C^ *p* = 0.051 ^C^
Systemic disease affecting the participant’s family member(s) Diabetes mellitus Coronary heart disease and atherosclerosis Hypertension Respiratory tract infections Not having any of the above diseases	155 (68.0%) 40 (53.3%) 133 (66.2%) 26 (68.4%) 117 (61.9%)	73 (32.0%) 35 (46.7%) 68 (33.8%) 12 (31.6%) 72 (38.1%)	*p* = 0.120 ^C^ *p* = 0.031 ^C^ *p* = 0.485 ^C^ *p* = 0.585 ^C^ *p* = 0.376 ^C^
Source of information Specialized person appearing in media Social media post by a specialized person General accounts on social media Scientific article My opinion I am in the dental field I am in the health field Personal message of a social group	11 (37.9%) 18 (40.0%) 32 (57.1%) 11 (35.5%) 199 (66.3%) 11 (64.7%) 15 (50.0%) 24 (61.5%)	18 (62.1%) 27 (60.0%) 24 (42.9%) 20 (64.5%) 101 (33.7%) 6 (35.3%) 15 (50.0%) 15 (38.5%)	*p* = 0.002 ^C^ *p* < 0.001 ^C^ *p* = 0.233 ^C^ *p* < 0.001 ^C^ *p* = 0.256 ^C^ *p* = 0.975 ^C^ *p* = 0.091 ^C^ *p* = 0.703 ^C^

^1^ Comparison test using a dichotomized scale with “agree” and “strongly agree” categories being merged; ^C^ chi-square test.

**Table 7 healthcare-11-00088-t007:** Multivariate analysis for the association between periodontal disease and diabetes mellitus.

Variable	Odds Ratio (95% CI)	*p*-Value
Dental health Untreated periodontal disease	1.96 (1.20 to 3.21)	*p* = 0.007
Systemic disease affecting the participant Diabetes mellitus	1.73 (0.99 to 3.04)	*p* = 0.056
Systemic disease affecting the participant’s family member(s) Coronary heart disease and atherosclerosis	1.50 (0.89 to 2.53)	*p* = 0.128
Source of information Specialized person appearing in media Social media post by a specialized person Scientific article	1.86 (0.81 to 4.30) 2.34 (1.19 to 4.57) 2.98 (1.33 to 6.68)	*p* = 0.145 *p* = 0.013 *p* = 0.008

**Table 8 healthcare-11-00088-t008:** Awareness about the association between periodontal disease and coronary heart disease and atherosclerosis distributed by various factors.

Variable	Strongly Disagree		Disagree		I Do not Know		Agree		Strongly Agree	
	*n*	(%)	*n*	(%)	*n*	(%)	*n*	(%)	*n*	(%)
Sex										
Male	8	3.2%	55	22.0%	134	53.6%	38	15.2%	15	6.0%
Female	11	4.4%	58	23.0%	126	50.0%	44	17.5%	13	5.2%
Age group										
18–24	7	7.5%	26	28%	42	45.2%	15	16.1%	3	3.2%
25–44	7	2.7%	54	20.8%	136	52.5%	45	17.4%	17	6.6%
45–64	5	3.6%	31	22.5%	75	54.3%	21	15.2%	6	4.3%
>65	0	0.0%	2	16.7%	7	58.3%	1	8.3%	2	16.7%
Education level										
Elementary school	0	0.0%	13	41.6%	12	37.5%	4	12.5%	3	9.4%
Middle school	2	6.1%	5	15.2%	18	54.5%	4	12.1%	4	12.1%
High school	3	4.7%	24	22.4%	54	50.5%	17	15.9%	7	6.5%
Post-secondary diploma	2	3.3%	11	18.0%	38	62.3%	10	16.4%	0	0.0%
Bachelor’s degree	9	3.8%	57	23.8%	121	50.6%	43	18.0%	9	3.8%
Master’s degree	0	0.0%	1	5.0%	13	65.0%	4	20.0%	2	10.0%
Ph.D.	1	10%	2	20.0%	4	40.0%	0	0.0%	3	30.0%
Frequency of dental visits										
Once every six months	4	5.9%	15	22.1%	30	44.1%	15	22.1%	4	5.9%
Once a year	1	1.8%	11	19.3%	33	57.9%	7	12.3%	5	8.8%
Only when I need dental treatment	6	3.3%	48	26.1%	89	48.4%	27	14.7%	14	7.6%
Only when I feel pain	6	3.7%	32	18.8%	91	56.2%	30	18.5%	3	1.9%
I never visit the dentist	2	6.5%	7	22.6%	17	54.8%	3	9.7%	2	6.5%
Dental health										
Untreated dental caries	7	3.2%	51	23.1%	120	54.3%	40	18.1%	3	1.4%
Untreated periodontal disease	3	3.4%	19	21.6%	37	42.0%	26	29.5%	3	3.4%
No dental caries or periodontal disease	11	4.6%	53	22.0%	124	51.5%	31	12.9%	22	9.1%
Systemic disease affecting the participant										
Diabetes mellitus	3	4.7%	25	39.1%	23	35.9%	9	14.1%	4	6.3%
Coronary heart disease and atherosclerosis	0	0.0%	7	26.9%	8	30.8%	10	38.5%	1	3.8%
Hypertension	1	1.4%	17	24.3%	35	50.0%	12	17.1%	5	7.1%
Respiratory tract infections	2	14.3%	3	21.4%	7	50.0%	2	14.3%	0	0.0%
Not having any of the above diseases	14	3.7%	79	21.0%	202	53.7%	61	16.2%	20	5.3%
Systemic disease affecting the participant’s family member(s)										
Diabetes mellitus	15	6.6%	57	25.0%	125	54.8%	29	12.7%	2	0.9%
Coronary heart disease and atherosclerosis	2	2.7%	18	24.0	30	40.0%	23	30.7%	2	2.7%
Hypertension	12	6.0%	46	22.9%	108	53.7%	30	14.9%	5	2.5%
Respiratory tract infections	1	2.6%	12	31.6%	16	42.1%	8	21.1%	1	2.6%
Not having any of the above diseases	1	0.5%	45	23.8%	92	48.7%	32	16.9%	19	10.1%
Source of information										
Specialized person appearing in written, visual, or audio media	3	10.3%	1	3.4%	10	34.5%	10	34.5%	5	17.2%
Post on social media by a specialized person	1	2.2%	4	8.9%	16	35.6%	21	46.7%	3	6.7%
General accounts on social media	2	3.6%	13	23.2%	19	33.9%	14	25.0%	8	14.3%
Scientific article	0	0.0%	3	9.7%	9	29.0%	13	41.9%	6	19.4%
My opinion	7	2.3%	78	26.0%	166	55.3%	43	14.3%	6	2.0%
I am in the dental field	1	5.9%	2	11.8%	10	58.8%	3	17.6%	1	5.9%
I am in the health field	2	6.7%	8	26.7%	12	40.0%	6	20.0%	2	6.7%
Personal message of a social group (for example WhatsApp, Telegram, …)	4	10.3%	9	23.1%	15	38.5%	10	25.6%	1	2.6%

The totals of percentages are more than 100% due to individuals having more than one condition simultaneously. Per-centages are calculated for each row.

**Table 9 healthcare-11-00088-t009:** Awareness about the association between periodontal disease and coronary heart disease and atherosclerosis distributed by various factors.

Variable	N (%) of Strongly Disagree, Disagree, Neutral	N (%) of Agree and Strongly Agree	Comparison Test ^1^
Sex Male Female	197 (78.8%) 195 (77.4%)	53 (21.2%) 58 (22.6%)	*p* = 0.701 ^C^
Age group 18–24 25–44 45–64 >65	75 (80.6%) 197 (76.1%) 111 (80.4%) 9 (75.0%)	18 (19.4%) 62 (23.9%) 27 (19.6%) 3 (25.0%)	*p* = 0.671 ^F^
Education level Elementary school Middle school High school Post-secondary diploma Bachelor’s degree Master’s degree Ph.D.	25 (78.1%) 25 (75.8%) 83 (77.6%) 51 (83.6%) 187 (78.2%) 14 (70.0%) 7 (70.0%)	7 (21.9%) 8 (24.2%) 24 (22.4%) 10 (16.4%) 52 (21.8%) 6 (30.0%) 3 (30.0%)	*p* = 0.841 ^F^
Frequency of dental visits Once every six months Once a year Only when I need dental treatment Only when I feel pain I never visit the dentist	49 (72.1%) 45 (78.9%) 143 (77.7%) 129 (79.6%) 26 (83.9%)	19 (27.9%) 12 (21.1%) 41 (22.3%) 33 (20.4%) 5 (16.1%)	*p* = 0.678 ^C^
Dental health Untreated dental caries Untreated periodontal disease No dental caries or periodontal disease	178 (80.5%) 59 (67.0%) 188 (78.0%)	43 (19.5%) 29 (33.0%) 53 (22.0%)	*p* = 0.238 ^C^ *p* = 0.006 ^C^ *p* = 0.967 ^C^
Systemic disease affecting the participant Diabetes mellitus Coronary heart disease and atherosclerosis Hypertension Respiratory tract infections Not having any of the above diseases	51 (79.7%) 15 (57.7%) 53 (75.7%) 12 (85.7%) 295 (78.5%)	13 (20.3%) 11 (42.3%) 17 (24.3%) 2 (14.3%) 81 (21.5%)	*p* = 0.740 ^C^ *p* = 0.010 ^C^ *p* = 0.605 ^C^ *p* = 0.744 ^F^ *p* = 0.729 ^C^
Systemic disease affecting the participant’s family member(s) Diabetes mellitus Coronary heart disease and atherosclerosis Hypertension Respiratory tract infections Not having any of the above diseases	197 (86.4%) 50 (66.7%) 166 (82.6%) 29 (76.3%) 138 (73.0%)	31 (13.6%) 25 (33.3%) 35 (17.4%) 9 (23.7%) 51 (27.0%)	*p* < 0.001 ^C^ *p* = 0.010 ^C^ *p* = 0.046 ^C^ *p* = 0.784 ^C^ *p* = 0.033 ^C^
Source of information Specialized person appearing in media Social media post by a specialized person General accounts on social media Scientific article My opinion I am in the dental field I am in the health field Personal message of a social group	14 (48.3%) 21 (46.7%) 34 (60.7%) 12 (38.7%) 251 (83.7%) 13 (76.5%) 22 (73.3%) 28 (71.8%)	15 (51.7%) 24 (53.3%) 22 (39.3%) 19 (61.3%) 49 (16.3%) 4 (23.5%) 8 (26.7%) 11 (28.2%)	*p* < 0.001 ^C^ *p* < 0.001 ^C^ *p* < 0.001 ^C^ *p* < 0.001 ^C^ *p* < 0.001 ^C^ *p* = 0.773 ^F^ *p* = 0.516 ^C^ *p* = 0.323 ^C^

^1^ Comparison test using a dichotomized scale with “agree” and “strongly agree” categories being merged; ^C^ chi-square test; ^F^ Fisher’s Exact test.

**Table 10 healthcare-11-00088-t010:** Multivariate analysis for the association between periodontal disease and coronary heart disease and atherosclerosis.

Variable	Odds Ratio (95% CI)	*p*-Value
Dental health Untreated periodontal disease	2.13 (1.20 to 3.77)	*p* = 0.010
Systemic disease affecting the participant Coronary heart disease and atherosclerosis	2.70 (1.36 to 5.36)	*p* = 0.004
Systemic disease affecting the participant’s family member(s) Diabetes mellitus Coronary heart disease and atherosclerosis Hypertension Not having any of the above diseases	0.28 (0.15 to 0.55) 2.79 (1.41 to 5.51) 0.81 (0.41 to 1.60) 0.92 (0.42 to 2.03)	*p* < 0.001 *p* = 0.003 *p* = 0.547 *p* = 0.843
Source of information Specialized person appearing in media Social media post by a specialized person General accounts on social media Scientific article My opinion	2.21 (0.90 to 5.45) 2.98 (1.45 to 6.15) 2.76 (1.38 to 5.52) 6.43 (2.64 to 15.67) 0.92 (0.54 to 1.56)	*p* = 0.085 *p* = 0.003 *p* = 0.004 *p* < 0.001 *p* = 0.754

**Table 11 healthcare-11-00088-t011:** Awareness about the association between periodontal disease and hypertension distributed by various factors.

Variable	Strongly Disagree		Disagree		I Do not Know		Agree		Strongly Agree	
	*n*	(%)	*n*	(%)	*n*	(%)	*n*	(%)	*n*	(%)
Sex										
Male	8	3.2%	34	13.6%	145	50.0%	47	18.8%	16	6.4%
Female	15	6.0%	33	13.1%	126	58.0%	58	23.0%	20	7.9%
Age group										
18–24	6	6.5%	16	17.2%	47	50.5%	20	21.5%	4	4.3%
25–44	10	3.9%	33	12.7%	144	55.6%	49	18.9%	23	8.9%
45–64	7	5.1%	17	12.3%	75	54.3%	33	23.9%	6	4.3%
>65	0	0.0%	1	8.3%	5	41.7%	3	25.0%	3	25.0%
Education level										
Elementary school	3	9.4%	5	15.6%	13	40.6%	9	28.1%	2	6.3%
Middle school	2	6.1%	4	12.1%	19	57.6%	4	12.1%	4	12.1%
High school	4	3.7%	21	19.6%	53	49.5%	18	16.8%	11	10.3%
Post-secondary diploma	3	4.9%	6	9.8%	35	57.4%	17	27.9%	0	0.0%
Bachelor’s degree	9	3.8%	28	11.7%	133	55.6%	54	22.6%	15	6.3%
Master’s degree	0	0.0%	1	5.0%	14	70.0%	3	15.0%	2	10.0%
Ph.D.	2	20.0%	2	20.0%	4	40.0%	0	0.0%	2	20.0%
Frequency of dental visits										
Once every six months	6	8.8%	10	14.7%	31	45.6%	17	25.0%	4	5.9%
Once a year	0	0.0%	10	17.5%	31	54.4%	12	21.1%	4	7.0%
Only when I need dental treatment	5	2.7%	28	15.2%	96	52.2%	37	20.1%	18	9.8%
Only when I feel pain	10	6.2%	18	11.1%	94	58.0%	31	19.1%	9	5.6%
I never visit the dentist	2	6.5%	1	3.2%	19	61.3%	8	25.8%	1	3.2%
Dental health										
Untreated dental caries	9	4.1%	27	12.2%	123	55.7%	49	22.2%	13	5.9%
Untreated periodontal disease	4	4.5%	8	9.1%	36	40.9%	36	40.9%	4	4.5%
No dental caries or periodontal disease	11	4.6%	36	14.9%	131	54.4%	41	17.0%	22	9.1%
Systemic disease affecting the participant										
Diabetes mellitus	4	6.3%	11	17.2%	29	45.3%	17	26.6%	3	4.7%
Coronary heart disease and atherosclerosis	2	7.7%	5	19.2%	10	38.5%	8	30.8%	1	3.8%
Hypertension	4	5.7%	7	10.0%	31	44.3%	24	34.3%	4	5.7%
Respiratory tract infections	1	7.1%	2	14.3%	7	50.0%	4	28.6%	0	0.0%
Not having any of the above diseases	13	3.5%	49	13.0%	214	56.9%	69	18.4%	31	8.2%
Systemic disease affecting the participant’s family member(s)										
Diabetes mellitus	13	5.7%	38	16.7%	123	53.9%	44	19.3%	10	4.4%
Coronary heart disease and atherosclerosis	4	5.3%	10	13.3%	31	41.3%	22	29.3%	8	10.7%
Hypertension	10	5.0%	25	12.4%	106	52.7%	51	25.4%	9	4.5%
Respiratory tract infections	0	0.0%	6	15.8%	21	55.3%	11	28.9%	0	0.0%
Not having any of the above diseases	5	2.6%	23	12.2%	103	54.5%	37	19.6%	21	11.1%
Source of information										
Specialized person appearing in written, visual, or audio media	4	13.8%	1	3.4%	9	31.0%	11	37.9%	4	13.8%
Post on social media by a specialized person	3	6.7%	2	4.4%	16	35.6%	14	31.1%	10	22.2%
General accounts on social media	3	5.4%	6	10.7%	20	35.7%	15	26.8%	12	21.4%
Scientific article	1	3.2%	3	9.7%	10	32.3%	11	35.5%	6	19.4%
My opinion	11	3.7%	43	14.3%	173	57.7%	60	20.0%	13	4.3%
I am in the dental field	0	0.0%	2	11.8%	10	58.8%	5	29.4%	0	0.0%
I am in the health field	2	6.7%	6	20.0%	11	36.7%	9	30.0%	2	6.7%
Personal message of a social group (for example WhatsApp, Telegram, …)	1	2.6%	4	10.3%	21	53.8%	12	30.8%	1	2.6%

The totals of percentages are more than 100% due to individuals having more than one condition simultaneously. Per-centages are calculated for each row.

**Table 12 healthcare-11-00088-t012:** Awareness about the association between periodontal disease and hypertension distributed by various factors.

Variable	N (%) of Strongly Disagree, Disagree, Neutral	N (%) of Agree and Strongly Agree	Comparison Test ^1^
Sex Male Female	187 (74.8%) 174 (69.0%)	63 (25.2%) 78 (31.0%)	*p* = 0.0152 ^C^
Age group 18–24 25–44 45–64 > 65	69 (74.2%) 187 (72.2%) 99 (71.7%) 6 (50.0%)	24 (25.8%) 72 (27.8%) 39 (28.3%) 6 (50.0%)	*p* = 0.385 ^F^
Education level Elementary school Middle school High school Post-secondary diploma Bachelor’s degree Master’s degree Ph.D.	21 (65.6%) 25 (75.8%) 78 (72.9%) 44 (72.1%) 170 (71.1%) 15 (75.0%) 8 (80.0%)	11 (34.4%) 8 (24.2%) 29 (27.1%) 17 (27.9%) 69 (28.9%) 5 (25.0%) 2 (20.0%)	*p* = 0.976 ^F^
Frequency of dental visits Once every six months Once a year Only when I need dental treatment Only when I feel pain I never visit the dentist	47 (69.1%) 41 (71.9%) 129 (70.1%) 122 (75.3%) 22 (71.0%)	21 (30.9%) 16 (28.1%) 55 (29.9%) 40 (24.7%) 9 (29.0%)	*p* = 0.827 ^C^
Dental health Untreated dental caries Untreated periodontal disease No dental caries or periodontal disease	159 (71.9%) 48 (54.5%) 178 (73.9%)	62 (28.1%) 40 (45.5%) 63 (26.1%)	*p* = 0.988 ^C^ *p* < 0.001 ^C^ *p* = 0.351 ^C^
Systemic disease affecting the participant Diabetes mellitus Coronary heart disease and atherosclerosis Hypertension Respiratory tract infections Not having any of the above diseases	44 (68.8%) 17 (65.4%) 42 (60.0%) 10 (71.4%) 276 (73.4%)	20 (31.3%) 9 (34.6%) 28 (40.0%) 4 (28.6%) 100 (26.6%)	*p* = 0.547 ^C^ *p* = 0.447 ^C^ *p* = 0.017 ^C^ *p* = 1.00 ^F^ *p* = 0.199 ^C^
Systemic disease affecting the participant’s family member(s) Diabetes mellitus Coronary heart disease and atherosclerosis Hypertension Respiratory tract infections Not having any of the above diseases	174 (76.3%) 45 (60.0%) 141 (70.1%) 27 (71.1%) 131 (69.3%)	54 (23.7%) 30 (40.0%) 60 (29.9%) 11 (28.9%) 58 (30.7%)	*p* = 0.045 ^C^ *p* = 0.013 ^C^ *p* = 0.473 ^C^ *p* = 0.902 ^C^ *p* = 0.314 ^C^
Source of information Specialized person appearing in media Social media post by a specialized person General accounts on social media Scientific article My opinion I am in the dental field I am in the health field Personal message of a social group	14 (48.3%) 21 (46.7%) 29 (51.8%) 14 (45.2%) 227 (75.7%) 12 (70.6%) 19 (63.3%) 26 (66.7%)	15 (51.7%) 24 (53.3%) 27 (48.2%) 17 (54.8%) 73 (24.3%) 5 (29.4%) 11 (36.7%) 13 (33.3%)	*p* = 0.004 ^C^ *p* < 0.001 ^C^ *p* < 0.001 ^C^ *p* < 0.001 ^C^ *p* = 0.023 ^C^ *p* = 1.00 ^F^ *p* = 0.281 ^C^ *p* = 0.448 ^C^

^1^ Comparison test using a dichotomized scale with “agree” and “strongly agree” categories being merged; ^C^ chi-square test; ^F^ Fisher’s Exact test.

**Table 13 healthcare-11-00088-t013:** Multivariate analysis for the association between periodontal disease and hypertension.

Variable	Odds Ratio (95% CI)	*p*-Value
Dental health Untreated periodontal disease	2.42 (1.45 to 4.03)	*p* < 0.001
Systemic disease affecting the participant Hypertension	1.52 (0.86 to 2.68)	*p* = 0.150
Systemic disease affecting the participant’s family member(s) Diabetes mellitus Coronary heart disease and atherosclerosis	0.55 (0.35 to 0.86) 1.99 (1.13 to 3.52)	*p* = 0.009 *p* = 0.018
Source of information Specialized person appearing in media Social media post by a specialized person General accounts on social media Scientific article My opinion	1.72 (0.74 to 4.02) 2.14 (1.07 to 4.26) 2.71 (1.44 to 5.11) 3.15 (1.37 to 7.25) 1.04 (0.65 to 1.67)	*p* = 0.208 *p* = 0.031 *p* = 0.002 *p* = 0.007 *p* = 0.865

**Table 14 healthcare-11-00088-t014:** Awareness about the association between periodontal disease and respiratory tract infections distributed by various factors.

Variable	Strongly Disagree		Disagree		I Do not Know		Agree		Strongly Agree	
	*n*	(%)	*n*	(%)	*n*	(%)	*n*	(%)	*n*	(%)
Sex										
Male	12	4.8%	43	17.2%	143	57.2%	38	15.2%	14	5.6%
Female	11	4.4%	56	22.2%	128	50.8%	44	17.5%	13	5.2%
Age group										
18–24	5	5.4%	20	21.5%	45	48.4%	19	20.4%	4	4.3%
25–44	9	3.5%	54	20.8%	143	55.2%	37	14.3%	16	6.2%
45–64	8	5.8%	23	16.7%	77	55.8%	25	18.1%	5	3.6%
>65	1	8.3%	2	16.7%	6	50.0%	1	8.3%	2	16.7%
Education level										
Elementary school	3	9.4%	7	21.9%	16	50.0%	4	12.5%	2	6.3%
Middle school	3	9.1%	4	12.1%	19	57.6%	4	12.1%	3	9.1%
High school	5	4.7%	24	22.4%	56	52.3%	17	15.9%	5	4.7%
Post-secondary diploma	2	3.3%	14	23.0%	34	55.7%	10	16.4%	1	1.6%
Bachelor’s degree	10	4.2%	44	18.4%	132	55.2%	40	16.7%	13	5.4%
Master’s degree	0	0.0%	3	15.0%	10	50.0%	6	30,.0%	1	5.0%
Ph.D.	0	0.0%	3	30.0%	4	40.0%	1	10.0%	2	20.0%
Frequency of dental visits										
Once every six months	5	7.4%	11	16.2%	33	48.5%	14	22.1%	4	5.9%
Once a year	0	0.0%	8	14.0%	33	57.9%	12	21.1%	4	7.0%
Only when I need dental treatment	6	3.3%	46	25.0%	87	47.3%	30	16.3%	15	8.2%
Only when I feel pain	11	6.8%	30	18.5%	96	59.3%	23	14.2%	2	1.2%
I never visit the dentist	1	3.2%	4	12.9%	22	71.0%	2	6.5%	2	6.5%
Dental health										
Untreated dental caries	12	5.4%	40	18.1%	123	55.7%	40	18.1%	6	2.7%
Untreated periodontal disease	5	5.7%	18	20.5%	42	47.7%	21	23.9%	2	2.3%
No dental caries or periodontal disease	9	3.7%	48	19.9%	132	54.8%	32	13.3%	20	8.3%
Systemic disease affecting the participant										
Diabetes mellitus	5	7.8%	18	28.1%	26	40.6%	14	21.9%	1	1.6%
Coronary heart disease and atherosclerosis	2	7.7%	4	15.4%	11	42.3%	7	26.9%	2	7.7%
Hypertension	6	8.6%	17	24.3%	32	45.7%	11	15.7%	4	5.7%
Respiratory tract infections	2	14.3%	4	28.6%	5	35.7%	3	21.4%	0	0.0%
Not having any of the above diseases	13	3.5%	71	18.9%	212	56.4%	59	15.7%	21	5.6%
Systemic disease affecting the participant’s family member(s)										
Diabetes mellitus	14	6.1%	54	23.7%	126	55.3%	33	14.5%	1	0.4%
Coronary heart disease and atherosclerosis	4	5.3%	13	17.3%	41	54.7%	16	21.3%	1	1.3%
Hypertension	16	8.0%	36	17.9%	112	55.7%	34	16.9%	3	1.5%
Respiratory tract infections	2	5.3%	7	18.4%	18	47.4%	11	28.9%	0	0.0%
Not having any of the above diseases	3	1.6%	36	19.0%	99	52.4%	29	15.3%	22	11.6%
Source of information										
Specialized person appearing in written, visual, or audio media	3	10.3%	2	6.9%	13	44.8%	9	31.0%	2	6.9%
Post on social media by a specialized person	2	4.4%	2	4.4%	24	53.3%	14	31.1%	3	6.7%
General accounts on social media	3	5.4%	7	12.5%	24	42.9%	13	23.2%	9	16.1%
Scientific article	0	0.0%	4	12.9%	14	45.2%	8	25.8%	5	16.1%
My opinion	10	3.3%	69	23.0%	168	56.0%	47	15.7%	6	2.0%
I am in the dental field	0	0.0%	4	23.5%	10	58.8%	3	17.6%	0	0.0%
I am in the health field	3	10.0%	5	16.7%	9	30.0%	11	36.7%	2	6.7%
Personal message of a social group (for example WhatsApp, Telegram, …)	2	5.1%	9	23.1%	19	48.7%	7	17.9%	2	5.1%

The totals of percentages are more than 100% due to individuals having more than one condition simultaneously. Percentages are calculated for each row.

**Table 15 healthcare-11-00088-t015:** Awareness about the association between periodontal disease and respiratory tract infections distributed by various factors.

Variable	N (%) of Strongly Disagree, Disagree, Neutral	N (%) of Agree and Strongly Agree	Comparison Test ^1^
Sex Male Female	198 (79.2%) 195 (77.4%)	52 (20.8%) 57 (22.6%)	*p* = 0.621 ^C^
Age group 18–24 25–44 45–64 >65	70 (75.3%) 206 (79.5%) 108 (78.3%) 9 (75.0%)	23 (24.7%) 53 (20.5%) 30 (21.7%) 3 (25.0%)	*p* = 0.815 ^F^
Education level Elementary school Middle school High school Post-secondary diploma Bachelor’s degree Master’s degree Ph.D.	26 (81.3%) 26 (78.8%) 85 (79.4%) 50 (82.0%) 186 (77.8%) 13 (65.0%) 7 (70.0%)	6 (18.7%) 7 (21.2%) 22 (20.6%) 11 (18.0%) 53 (22.2%) 7 (35.0%) 3 (30.0%)	*p* = 0.761 ^F^
Frequency of dental visits Once every six months Once a year Only when I need dental treatment Only when I feel pain I never visit the dentist	49 (72.1%) 41 (71.9%) 139 (75.5%) 137 (84.6%) 27 (87.1%)	19 (27.9%) 16 (28.1%) 45 (24.5%) 25 (15.4%) 4 (12.9%)	*p* = 0.064 ^C^
Dental health Untreated dental caries Untreated periodontal disease No dental caries or periodontal disease	175 (79.2%) 65 (73.9%) 189 (78.4%)	46 (20.8%) 23 (26.1%) 52 (21.6%)	*p* = 0.665 ^C^ *p* = 0.268 ^C^ *p* = 0.943 ^C^
Systemic disease affecting the participant Diabetes mellitus Coronary heart disease and atherosclerosis Hypertension Respiratory tract infections Not having any of the above diseases	49 (76.6%) 17 (65.4%) 55 (78.6%) 11 (78.6%) 296 (78.7%)	15 (23.4%) 9 (34.6%) 15 (21.4%) 3 (21.4%) 80 (21.3%)	*p* = 0.720 ^C^ *p* = 0.101 ^C^ *p* = 0.950 ^C^ *p* = 1.00 ^F^ *p* = 0.682 ^C^
Systemic disease affecting the participant’s family member(s) Diabetes mellitus Coronary heart disease and atherosclerosis Hypertension Respiratory tract infections Not having any of the above diseases	194 (85.1%) 58 (77.3%) 164 (81.6%) 27 (71.1%) 138 (73.0%)	34 (14.9%) 17 (22.7%) 37 (18.4%) 11 (28.9%) 51 (27.0%)	*p* < 0.001 ^C^ *p* = 0.828 ^C^ *p* = 0.142 ^C^ *p* = 0.261 ^C^ *p* = 0.026 ^C^
Source of information Specialized person appearing in media Social media post by a specialized person General accounts on social media Scientific article My opinion I am in the dental field I am in the health field Personal message of a social group	18 (62.1%) 28 (62.2%) 34 (60.7%) 18 (58.1%) 247 (82.3%) 14 (82.4%) 17 (56.7%) 30 (76.9%)	11 (37.9%) 17 (37.8%) 22 (39.3%) 13 (41.9%) 53 (17.7%) 3 (17.6%) 13 (43.3%) 9 (23.1%)	*p* = 0.029 ^C^ *p* = 0.006 ^C^ *p* < 0.001 ^C^ *p* = 0.005 ^C^ *p* = 0.007 ^C^ *p* = 1.00 ^F^ *p* = 0.003 ^C^ *p* = 0.830 ^C^

^1^ Comparison test using a dichotomized scale with “agree” and “strongly agree” categories being merged; ^C^ chi-square test; ^F^ Fisher’s Exact test.

**Table 16 healthcare-11-00088-t016:** Multivariate analysis for the association between periodontal disease and respiratory tract infections.

Variable	Odds Ratio (95% CI)	*p*-Value
Systemic disease affecting the participant’s family member(s) Diabetes mellitus Not having any of the above diseases	0.45 (0.24 to 0.84) 0.97 (0.53 to 1.77)	*p* = 0.012 *p* = 0.924
Source of information Specialized person appearing in media Social media post by a specialized person General accounts on social media Scientific article My opinion I am in the health field	1.68 (0.69 to 4.06) 1.73 (0.85 to 3.52) 2.99 (1.54 to 5.78) 2.62 (1.12 to 6.13) 0.99 (0.60 to 1.64) 3.71 (1.64 to 8.39)	*p* = 0.251 *p* = 0.133 *p* = 0.001 *p* = 0.026 *p* = 0.975 *p* = 0.002

## Data Availability

Data will be available upon request from the corresponding author.

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
