# Peer review of "Public Awareness of the Association between Periodontal Disease and Systemic Disease"

_healthcare, 2022, doi:10.3390/healthcare11010088_

Round 1
Reviewer 1 Report
A study was organised to assess of general population awareness on the association between Periodontal disease with selected non-communicable diseases (i.e., Diabetes mellitus, coronary heart disease and atherosclerosis, hypertension, and respiratory tract infections in Saudi Arabia. Some 502 participants responded a computer-assisted survey. The overall conclusion of the study was that there was a low level of awareness of public of these conditions.
This is an important topic, and I congratulate the authors for choosing it. However, on its current forms there are several major points that need to be addressed.
Methods
This is a large study with a very simple, descriptive only analysis.
I am also concerned in the sample size. Although the sample size seems to be large enough, there is no justification for its size, no calculations are reported, etc. Why 50 participants on each region? The manuscript should include a discussion of the power achieved to make comparisons.
Results
The results section only provides description of the results. There is no testing of associations, multivariate analysis, etc. That analysis would have been of value. A multivariate analysis would have allowed for the assessment of the relationship of the dependent variable under study (e.g., awareness) with the independent variables (e.g., level of education, sex, age, region, etc.).
Data is ordinal; however, the result present mean and standard deviation (Table 3). These measures are not the best to describe ordinal data.
Discussion
Although I generally agree with the discussion, but it reads more like a literature review. Only marginally relates to the results presented. It should go beyond background information to what is already known from the literature.
There is no multivariate analysis, , or at least bivariate analysis. Without this analysis, the conclusions may go beyond the data and results presented. For example, it is not possible to conclude about the effect of education, sex, etc.
Minor points.
Sex or gender?
Lines 131 to 134 repeats text from lines 125-128
Author Response
Reviewer 1 comments:
Thank you for taking the time to review our manuscript, we appreciate your comments that improved its quality.
Comment:
A study was organised to assess of general population awareness on the association between Periodontal disease with selected non-communicable diseases (i.e., Diabetes mellitus, coronary heart disease and atherosclerosis, hypertension, and respiratory tract infections in Saudi Arabia. Some 502 participants responded a computer-assisted survey. The overall conclusion of the study was that there was a low level of awareness of public of these conditions.
This is an important topic, and I congratulate the authors for choosing it. However, on its current forms there are several major points that need to be addressed.
Response:
Thank you
Comment:
This is a large study with a very simple, descriptive only analysis.
I am also concerned in the sample size. Although the sample size seems to be large enough, there is no justification for its size, no calculations are reported, etc. Why 50 participants on each region? The manuscript should include a discussion of the power achieved to make comparisons.
Response:
Sample size calculation method was added L. 72-73
Comment:
- The results section only provides description of the results. There is no testing of associations, multivariate analysis, etc. That analysis would have been of value. A multivariate analysis would have allowed for the assessment of the relationship of the dependent variable under study (e.g., awareness) with the independent variables (e.g., level of education, sex, age, region, etc.).
- Data is ordinal; however, the result present mean and standard deviation (Table 3). These measures are not the best to describe ordinal data.
Response:
A- As per your recommendation, inferential statistical analysis was performed with bivariate and multivariate analysis, methods (L. 117-127), results, discussion, and conclusion sections were modified accordingly
B- Means and standard deviation were deleted from (Table 3)
Comment:
Discussion
A- Although I generally agree with the discussion, but it reads more like a literature review. Only marginally relates to the results presented. It should go beyond background information to what is already known from the literature.
B- There is no multivariate analysis, , or at least bivariate analysis. Without this analysis, the conclusions may go beyond the data and results presented. For example, it is not possible to conclude about the effect of education, sex, etc.
Response:
- The discussion section was revised and updated.
- As per your recommendation, inferential statistical analysis was performed with bivariate and multivariate analysis, methods (L117-127), results, discussion, and conclusion sections were modified accordingly
Comment:
Minor points.
Sex or gender?
Lines 131 to 134 repeats text from lines 125-128
Response:
- Gender was changed to Sex throughout the manuscript.
- Lines 131 to 134 were deleted
Reviewer 2 Report
The authors aimed to assess the general public's awareness regarding the connection between PD and NCDs, mainly DM, CHDA, HT, and respiratory disease.
The study covers some issues that have been overlooked in other similar topics. The structure of the manuscript appears adequate and well divided in the sections. Moreover, the study is easy to follow, but some issues should be improved. Some of the comments that would improve the overall quality of the study are:
a. Authors must pay attention to the technical terms acronyms they used in the text;
b. Limitations of the study needs to be added;
c. Conclusion Section: please improve it, including some "take-home message".
Author Response
Reviewer 2 comments:
Thank you for taking the time to review our manuscript, we appreciate your comments that improved its quality.
Comment:
The authors aimed to assess the general public's awareness regarding the connection between PD and NCDs, mainly DM, CHDA, HT, and respiratory disease.
The study covers some issues that have been overlooked in other similar topics. The structure of the manuscript appears adequate and well divided in the sections. Moreover, the study is easy to follow, but some issues should be improved. Some of the comments that would improve the overall quality of the study are:
Response:
Thank you
Comment:
- Authors must pay attention to the technical terms acronyms they used in the text;
Response:
Thanks for highlighting this issue, we revised the manuscript to show the correct acronyms. In addition, we decreased the use of acronyms whenever possible.
Comment:
- Limitations of the study needs to be added;
Response:
Limitations of this study were added in the discussion L 381-384
Comment:
- Conclusion Section: please improve it, including some "take-home message".
Response:
Take home message was added to the end of the conclusion L448-451
Reviewer 3 Report
The paper of Alsalleh et at is about assessing public awareness about the connection between periodontal diseases and systematic conditions. Overall, the paper is scientifically sound but it needs further improvement in terms of adding more details about the awareness among health care professionals. The attached pdf includes comments that may help the authors improve their manuscript

Author Response
Reviewer 3 comments:
Thank you for taking the time to review our manuscript, we appreciate your carefully placed comments on the PDF file that improved its quality.
Comment:
The paper of Alsalleh et at is about assessing public awareness about the connection between periodontal diseases and systematic conditions. Overall, the paper is scientifically sound but it needs further improvement in terms of adding more details about the awareness among health care professionals. The attached pdf includes comments that may help the authors improve their manuscript
Response:
Thank you
Comment:
please mention few words about the survey, for example, how many questions? open or closed questions?
Response:
The following sentence was added to the survey design section:
The survey had ten close-ended questions (L 105-106).
In addition, a copy of the survey was attached in the supplementary files.
Comment:
Public awareness in any country is technically based on the professional awareness. Although people can learn from TV and socialmedia, the most reliable source for information is from medical professional.
Therefore, the authors need to add details about the level of awareness regarding the connection between periodontal diseases and other systematic diseases among health care professionals. These details should be included in both the introduction and the discussion parts. Of course, the cited papers must be conducted in Saudi Arabia in particular. For this purpose, the authors need to cite and consider the following papers:
DOI: 10.15537/smj.2018.11.23267
doi.org/10.1371/journal.pone.0276479
DOI: 10.1159/000495881
So, based on the information in these papers, we can know whether getting low public awareness is due to the lack of the professional awareness or due to other reasons
Response:
We would like to thank you for bringing this up. We have modified the introduction to reflect what is stated in the papers, as well as the discussion.
In the introduction L 56 to 58 as follows “However, while health care providers have a high level of awareness regarding the association between periodontal disease and systemic diseases [20–23], patients, on the other hand, have low levels of awareness regarding this link [24].”
In the discussion L 426 to 432 “Although the members of the medical and dental health team have high level of awareness regarding periodontal disease association with systemic diseases [20–22], this knowledge is yet to be transferred effectively to patients as demonstrated in this and previous studies [24]. It seems that lack of time, inadequate knowledge and training in oral health as well as lack of interaction with oral health care providers are the main barriers to providing patient education about the association between oral and systemic disease [21].”
Comment:
please cite more relevant papers that defined periodontal diseases
Response:
The definition was revised, and a more relevant paper cited
Comment:
please cite more relevant paper. The cited one must be either original research or meta-analysis
Response:
A systematic review with meta-analysis was cited
- Teshome, A.; Yitayeh, A. The Effect of Periodontal Therapy on Glycemic Control and Fasting Plasma Glucose Level in Type 2 Diabetic Patients: Systematic Review and Meta-Analysis. BMC Oral Health 2016, 17, 31, doi:10.1186/s12903-016-0249-1.
Comment:
in the previous paragraph (lines 47 to 51), the authors mentioned that patients have low levels of 50 awareness, Therefore, we cannot say that the evidence is limited, unless if the authors specifically talking about Saudi Arabia. please clarify this point
Response:
Most of the studies are conducted on patients as stated in our text “However, in general, patients have low levels of awareness regarding this link”
In line 54 and 55 we are talking about the general public not patients per se “Limited evidence in the literature is available addressing the general public's awareness regarding the connection between oral and systemic disease”
Comment:
please attach a copy of this survey in the supplementary files
Response:
Done, a copy of the survey was attached in the supplementary files.
Comment:
please try to minimize using abbreviations as much as possible, at his stage, these abbreviations are very confusing and readers need to go back and forth several times within the text to find the meaning of some abbreviations. For example, I understand that "coronary heart disease and atherosclerosis" is long so you used an abbreviation, but for other terms like dental caries, peridontal disease hypertension, diabetes, there is no need for these abbreviations
Response:
We have done the abbreviations as per the instructions for authors of the journal “Acronyms/Abbreviations/Initialisms should be defined the first time they appear in each of three sections: the abstract; the main text; the first figure or table. When defined for the first time, the acronym/abbreviation/initialism should be added in parentheses after the written-out form”, however, changes were made as suggested in the results and discussion section (excluding the tables).
However, we revised the manuscript and decreased the use of acronyms whenever possible.
Comment:
adding the levels of likert scale to the table would be better than using letters
Response:
Done for Tables 3, 5, 6, 7, and 8
Comment:
same as table 3
Response:
Done for Tables 3, 5, 6, 7, and 8
Comment:
the current discussion needs further improvement. the discussion should discuss and justify results rather than only saying that our results are similar to X and Y.
Response:
The discussion section was revised and updated.
Comment:
at this stage readers do not know anything about the awareness level among health care professionals or the content of medical curriculum, to avoid providing speculation, awareness among health care professionals must be added. Please see my previous comment in the introduction part
Response:
We would like to thank you for bringing this up. We have modified the introduction to reflect what is stated in the papers, as well as the discussion.
In the introduction L 56 to 58 as follows “However, while health care providers have a high level of awareness regarding the association between periodontal disease and systemic diseases [20–23], patients, on the other hand, have low levels of awareness regarding this link [24].”
In the discussion L 426 to 432 “Although the members of the medical and dental health team have high level of awareness regarding periodontal disease association with systemic diseases [20–22], this knowledge is yet to be transferred effectively to patients as demonstrated in this and previous studies [24]. It seems that lack of time, inadequate knowledge and training in oral health as well as lack of interaction with oral health care providers are the main barriers to providing patient education about the association between oral and systemic disease [21].”
Comment:
This site can’t be reached, please update the link
Response:
We checked the link to the websites, and it is working. However, for cybersecurity reasons some websites in Saudi Arabia do not allow access from some countries.
Comment:
This site can’t be reached, please update the link
Response:
We checked the link to the websites, and it is working. However, for cybersecurity reasons some websites in Saudi Arabia do not allow access from some countries.

Round 2
Reviewer 3 Report
The authors responded to the comments and improved their manuscript appropriately. I would recommend accepting in the current form
Thanks